# Imprints of Lockdown and Treatment Processes on the Wastewater Surveillance of SARS-CoV-2: A Curious Case of Fourteen Plants in Northern India

**Sudipti Arora** [1,*,†], **Aditi Nag** [1,†], **Ankur Rajpal** [2], **Vinay Kumar Tyagi** [2,*], **Satya Brat Tiwari** [3], **Jasmine Sethi** [1], **Devanshi Sutaria** [1], **Jayana Rajvanshi** [1], **Sonika Saxena** [1], **Sandeep Kumar Shrivastava** [4], **Vaibhav Srivastava** [5], **Akhilendra Bhushan Gupta** [6], **Absar Ahmed Kazmi** [2] and **Manish Kumar** [7]

1   Dr. B. Lal Institute of Biotechnology, 6-E, Malaviya Industrial Area, Malviya Nagar, Jaipur 302017, India; aditinag1@gmail.com (A.N.); sethi.28.jasmine@gmail.com (J.S.); devanshisnehal@gmail.com (D.S.); rajvanshi97@gmail.com (J.R.); sonika@blalbiotech.com (S.S.)
2   Environmental Biotechnology Group (EBiTG), Department of Civil Engineering, Indian Institute of Technology, Roorkee 247667, India; ankur.envt@gmail.com (A.R.); kazmifce@iitr.ac.in (A.A.K.)
3   School of Civil and Environmental Engineering, Nanyang Technological University, Singapore 637141, Singapore; sbtiwarigroup@gmail.com
4   Centre for Innovation, Research & Development (CIRD), Dr. B. Lal Clinical Laboratory Pvt. Ltd., Jaipur 302017, India; sandeepshrivastava@blallab.com
5   Discipline of Earth Science, Indian Institute of Technology Gandhinagar, Gujarat 382-355, India; vaibhavsri921@gmail.com
6   Department of Civil Engineering, Malaviya National Institute of Technology, Jaipur 302017, India; abgupta.ce@mnit.ac.in
7   School of Engineering, University of Petroleum & Energy Studies, Dehradun 248007, India; manish.kumar@ddn.upsc.ac.in
*   Correspondence: sudiptiarora@gmail.com or sudiptiarora@blalbiotech.com (S.A.); vinayiitrp@gmail.com or vinay.tyagi@ce.iitr.ac.in (V.K.T.); Tel.: +91-9829675677 (S.A.)
†   Sudipti Arora and Aditi Nag are equal contributors and thus shared the first authorship.

**Abstract:** The present study investigated the detection of severe acute respiratory syndrome–coronavirus 2 (SARS-CoV-2) genomes at each treatment stage of 14 aerobic wastewater treatment plants (WWTPs) serving the major municipalities in two states of Rajasthan and Uttarakhand in Northern India. The untreated, primary, secondary and tertiary treated wastewater samples were collected over a time frame ranging from under-lockdown to post-lockdown conditions. The results showed that SARS-CoV-2 RNA was detected in 13 out of 40 wastewater samples in Jaipur district, Rajasthan and in 5 out of 14 wastewater samples in the Haridwar District, Uttarakhand with the E gene predominantly observed as compared to the N and RdRp target genes in later time-points of sampling. The Ct values of genes present in wastewater samples were correlated with the incidence of patient and community cases of COVID-19. This study further indicates that the viral RNA could be detected after the primary treatment but was not present in secondary or tertiary treated samples. This study implies that aerobic biological wastewater treatment systems such as moving bed biofilm reactor (MBBR) technology and sequencing batch reactor (SBR) are effective in virus removal from the wastewater. This work might present a new indication that there is little to no risk in relation to SARS-CoV-2 while reusing the treated wastewater for non-potable applications. In contrast, untreated wastewater might present a potential route of viral transmission through WWTPs to sanitation workers and the public. However, there is a need to investigate the survival and infection rates of SARS-CoV-2 in wastewater.

**Keywords:** aerobic wastewater treatment; COVID-19; RT-qPCR based detection; SARS-CoV-2; sewage surveillance; wastewater based epidemiology

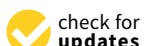

## 1. Introduction

The coronavirus disease (COVID-19) caused by Severe Acute Respiratory Syndrome Coronavirus 2 (SARS-CoV-2) emerged as a worldwide public health emergency within a few months of its outbreak in Wuhan, China in 2019. The extent of the pandemic COVID-19 is widespread and is currently confirmed to be present in more than 213 countries/regions worldwide [1,2]. SARS-CoV-2 spreads through air droplets and physical contact [3]. Hence, early detection and rapid containment protocols are crucial for its control and elimination. It is challenging to check and control the disease spread in developing countries like India because of the dense population [3]. These challenges are evidenced by an initial gradual increase in infection rates during the lockdown, followed by a sharp increase in the number of positive cases alongside the lifting of lockdown restrictions in India. In India, the number of COVID-19 positive cases increased from 1993 in May 2020 to 78,512 in August 2020, as of the last day of August 2020 [4] due to the lifting of the lockdown in early June 2020, signifying the immediate need of attention. Wastewater-based epidemiology (WBE) is a useful tool for community-wide detection of epidemics and pandemics in a given community.

The combination of clinical and environmental surveillance has been useful for public health practices [5]. Relying on clinical testing alone for detection and control is insufficient due to the scale of the spread and the existence of many asymptomatic and pauci-symptomatic cases. However, current evidence indicates that there is a need for better understanding of the role of wastewater as a potential source of epidemiological data and as a factor of public health risk. A well-validated WBE is imperative for viral surveillance, while appropriate sampling, the concentration of the virus in wastewater, population dynamics, and ethical concerns are crucial factors for a reliable WBE approach, particularly in regard to its utility as an early warning system [6].

The third quarter of 2020 has been exceptional in terms of discovering new knowledge pertaining to the WBE of SARS-CoV-2 genetic material loads, its testing methods, and various strong implications emerging from its use around the world. SARS-CoV-2 RNA was reported in wastewater in Brazil even before the first clinically confirmed cases [7], and various other studies detected SARS-CoV-2 RNA in wastewater across the globe in untreated wastewater [8–12]. However, there are very few recent studies on the occurrence of SARS-CoV-2 throughout the entire process of wastewater treatment in wastewater treatment plants (WWTP); which highlights the removal of the SARS-CoV-2 genome around the globe, such as in Spain [13,14], USA [15], Japan [16], Italy [17], Germany [18], Paris [19], China [20] and India [6,21–25]. Some of these studies have reported the presence of SARS-CoV-2 even in treated wastewater, and the main findings are highlighted in Table 1. In particular, in the study undertaken by Wurtzer et al. [19], six samples of treated wastewater were found to be positive for SARS-CoV-2 [19]. Randazzo et al. [14] found secondary treated samples to be positive, while none of the tertiary treated effluents was positive. Westhaus et al. [18] reported positive samples even after tertiary treatment (ozonation). It is important to investigate the presence of the virus in wastewater, as the treated effluent is to be utilized for irrigation purposes.

The study from India by Kumar et al. [7] reported two-point samplings, taken on 8 May 2020 and 27 May 2020, regarding anaerobic treatment systems (UASB). Thus, there remain questions pertaining to the capability of aerobic WWTPs from India at each stage of the treatment to achieve the decay of SARS-CoV-2 RNA in wastewater. Although WBE-based surveillance is a systematically integrated technique used for detection and diagnostics in many countries worldwide [26], India still lacks awareness about the far-reaching benefits of this surveillance. While infrastructure development is quintessential and a long-drawn process, sufficient good quality data on WBE from these regions can be useful for future planning and computational modeling. Furthermore, there are challenges and apprehensions about implementing WBE in developing countries, such as India, due to the poor water supply network and sewerage system. Thus, the present study is important in the context of WBE from an Indian perspective around its capability, owing to

unplanned and incomplete sewer systems, sewer overflows and leakage situations, and strong seasonal fluctuations. These considerations warranted a study that can track the presence of SARS-CoV-2 viral RNA after each of the wastewater treatment stages in Indian settings in order to understand the effects of wastewater treatment on RNA decay. Hence, this study will help to allay the commonly-perceived fear of the commons pertaining to the effectiveness of WWTPs.

**Table 1.** Review on the efficacy of treatment processes in different stages of wastewater treatment plants (WWTPs) for the removal of SARS-CoV-2 RNA in different studies.

| Country | No. of WWTP | Treatment Stages | Process Used | Result | References |
|---|---|---|---|---|---|
| Spain | 6 | Untreated | - | Positive samples: 35/42 | [14] |
| | | Secondary treatment | Activated sludge | Positive samples: 2/18 | |
| | | Tertiary treatment | Coagulation, Flocculation, Sand filtration, Disinfection (UV, NaClO) | Positive samples: 0/12 | |
| Spain | 1 | Primary treated | Primary settler | Positive samples: 1/4 | [13] |
| | | Secondary treatment and Tertiary treatment | SBR and Chemical removal, Microfiltration | Positive samples: 0/5 | |
| Southern Louisiana, USA | 2 | Untreated | - | Positive samples: 2/7 | [15] |
| | | Secondary treated | Activated sludge | Positive samples: 0/4 | |
| | | Tertiary treated | Chlorination | Positive samples: 0/4 | |
| Japan | 1 | Untreated | - | Positive samples: 0/5 | [16] |
| | | Secondary treated | Activated sludge and aeration | Positive samples: 1/5 | |
| | | River water | - | Positive samples: 0/3 | |
| Italy | 3 | Untreated | - | Positive samples: 4/8 | [17] |
| | | Tertiary treated | Peracetic acid or high-intensity UV lamps | Positive samples: 0/4 | |
| Germany | 9 | Untreated | Activated sludge | Positive samples: 9/9 | [19] |
| | | Tertiary treated | Ozonation | Positive samples: 4/4 | |
| France | 3 | Untreated wastewater | - | Positive samples: 23/23 | [18] |
| | | Tertiary treated | Data not available | Positive samples: 6/8 | |
| China | 1 | Untreated | - | Positive samples: 0/4 | [20] |
| | | Tertiary treated | Septic tank | Positive samples: 7/9 | |
| India | 1 | Untreated wastewater | - | Positive samples: 1/2 | [22] |
| | | Primary treated | UASB Inlet | Positive samples: 1/2 | |
| | | Secondary treated | UASB Outlet | Positive samples: 1/2 | |
| | | Tertiary treated | Aeration | Positive samples: 1/2 | |
| | | Final Effluent | - | Positive samples: 0/2 | |
| India | 14 | Untreated and primary treated | - | Positive samples: 12/33 | Present study |
| | | Secondary treated | MBBR, SBR | Positive samples: 0/7 | |
| | | Tertiary treated | UV Chlorination | Positive samples: 0/14 | |

Note: - = Information not available, MBBR= Moving Bed biofilm reactor, NaClO = Sodium hypochlorite, SBR = Sequencing Batch Reactor, $Cl_2$ = Chlorine disinfection, UV—Ultra violet disinfection, MLD = million liters per day, UASB = Up-flow Anaerobic Sludge Blanket.

Recently, the world Sustainable Development Summit 2021 highlighted the importance of WBE for SARS-CoV-2 monitoring and discussed the various challenges involved in implementing WBE in India, in order to create policy making decisions. In this context, the main objective of this study is to track the aerobic biological wastewater treatment system for the decay of SARS-CoV-2 and its genomic RNA along the treatment process and evaluate its effectiveness. This study was conducted to detect SARS-CoV-2 RNA

in both untreated and treated wastewater samples collected from multiple locations in order to assess the health risks posed by the reuse of effluents coming from WWTPs. Additionally, an attempt has been made to correlate the detected Threshold Cycle (Ct) values of target genes viz. RNA-dependent polymerase (RdRP) gene, nucleocapsid (N) SARS-CoV-2 specific genes, and gene E, which are characteristic of pan-Sarbecoviruses in the qualitative detection of SARS-CoV-2. This can serve as a crude indication of the genome load over the time-frame which includes lockdown, partial lockdown and no lockdown conditions. The experiments were carried out to detect the presence of SARS-CoV-2 in influent, primary, secondary, and tertiary treated effluent samples from 14 wastewater treatment systems (13 WWTPs and one pump house) in four cities (Roorkee, Rishikesh, Haridwar, Jaipur) in the two North Indian states of Uttarakhand and Rajasthan, and to possibly decipher the potential of current biological treatment systems for removing the virus. This is the study report for a comprehensive data analysis that gives insights into aerobic biological treatment systems' role in decaying the SARS-CoV-2 viral genome.

## 2. Material and Methods

### 2.1. Sample Collection and Transportation

The wastewater samples (grab samples) (1 L) were collected during nine different time-points from seven wastewater treatment facilities in Jaipur city (Rajasthan), and grab and composite samples were collected during three different time points from seven wastewater treatment facilities in Uttarakhand state (from the cities of Rishikesh, Haridwar, and Roorkee). The samples were collected during the morning hours, between 6 AM and 10 AM, from different wastewater treatment systems when the sewage flow rate was higher. The samples collected in Uttarakhand were transported through cold chain transportation to Jaipur. All samples were pre-processed in the Environmental Biotechnology Laboratory at Dr. B. Lal Institute of Biotechnology, Jaipur. It was ensured during the packaging that the samples stayed at 4 °C for the whole transit. After sampling, the sampling bottle's surface was disinfected with 90% ethanol, was labeled, and was immediately transported (2–4 °C) to the laboratory. The sampling was carried out during the months of May to August 2020. Detailed information on the numbers of samples, sampling sites, designed capacity, the average flow rate of WWTPs, and the current treatment (secondary and tertiary) technology from the states of Rajasthan and Uttarakhand is provided in Table 2. The locations sites of the different WWTPs are highlighted in Figure 1.

**Table 2.** Details on the sampling location sites along with treatment characteristics of WWTPs located in Rajasthan and Uttarakhand states.

| Site No. | Sampling Location | Type of Secondary Treatment Technology | Type of Tertiary Treatment | Dosage &Contact Time of Tertiary Treatment | Design Capacity (MLD) | Flow Rate (Avg. MLD) | Number of Connected Residents (Approx.) |
|---|---|---|---|---|---|---|---|
| Site 1 | Ramniwas Garden, Jaipur 26.8963° N, 75.8100° E | MBBR | UV | NA | 1 | ~1 | >7000 |
| Site 2 | Central Park, Jaipur 26.9048° N, 75.8073° E | SBR | Cl₂ (Bleach Powder) | 4 ppm by dropping system | 1 | ~1 | >7000 |
| Site 3 | Delawas, Jaipur 27.3735° N, 75.8926° E | ASP | No treatment | 3 ppm, 30 min | 65 | ~62.5 | >480,000 |
| Site 4 | Jawahar Circle, Jaipur 26°50′29″ N,75°48′0″ E | MBBR | UV | NA | 1 | ~1 | >7000 |
| Site 5 | Brahmpuri, Jaipur 26.9373° N, 75.8250° E | SBR | No treatment | NA | 27MLD | ~8 | >59,000 |
| Site 6 | MNIT, Jaipur 26.8640° N, 75.8108° E | MBBR | Cl₂ (Hypochlorite) | 2.5–3 ppm, 30 min | 1 | ~1 | >7000 |
| Site 7 | Dravyavati River Project, Jaipur 26.7980° N, 75.8039° E | SBR | Cl₂ (Hypochlorite) | 3–5 ppm, 30 min | 65 | ~65 | >480,000 |
| Site 8 | IIT Roorkee 29.8649° N, 77.8965° E | SBR | UV | 40000 microwatt sec/cm² | 3 | ~1.5 | >7000 |

**Table 2.** *Cont.*

| Site No. | Sampling Location | Type of Secondary Treatment Technology | Type of Tertiary Treatment | Dosage &Contact Time of Tertiary Treatment | Design Capacity (MLD) | Flow Rate (Avg. MLD) | Number of Connected Residents (Approx.) |
|----------|-------------------|------------------------------------------|-----------------------------|-----------------------------------------------|-------------------------|------------------------|-----------------------------------------|
| Site 9 | Muni kiReti, Rishkesh 30.1199° N, 78.3031° E | MBBR | $Cl_2$ (Hypochlorite) | 3 ppm, 30 min | 5 | ~3.5 | >15,000 |
| Site 10 | Swarg Ashram, Rishikesh 30.1165° N, 78.3131° E | SBR | $Cl_2$ (Hypochlorite) | 3 ppm, 30 min | 3 | ~1.5 | >10,000 |
| Site 11 | Chandreshwar Nagar, Rishikesh 30.1115° N, 78.3056° E | MBBR | $Cl_2$ (Hypochlorite) | 3 ppm, 30 min | 7.5 | ~2.5 | >10,000 |
| Site 12 | Sarai, Haridwar 29.9043° N, 78.1080° E | SBR | $Cl_2$ (Hypochlorite) | 3 ppm, 30 min | 14 | ~15 | >100,000 |
| Site 13 | Jagjeetpur, Haridwar 29.9174° N, 78.1316° E | MBBR | $Cl_2$ (Hypochlorite) | 3 ppm, 30 min | 68 | ~45 | >200,000 |
| Site 14 | Pump House, Haridwar | NA | NA | NA | NA | NA | >200,000 |

Note: NA = Information not available, ASP = Activated sludge process, MBBR = Moving Bed biofilm reactor, SBR = Sequencing Batch Reactor, $Cl_2$ = Chlorine disinfection, UV—Ultra violet disinfection, MLD = million liters per day.

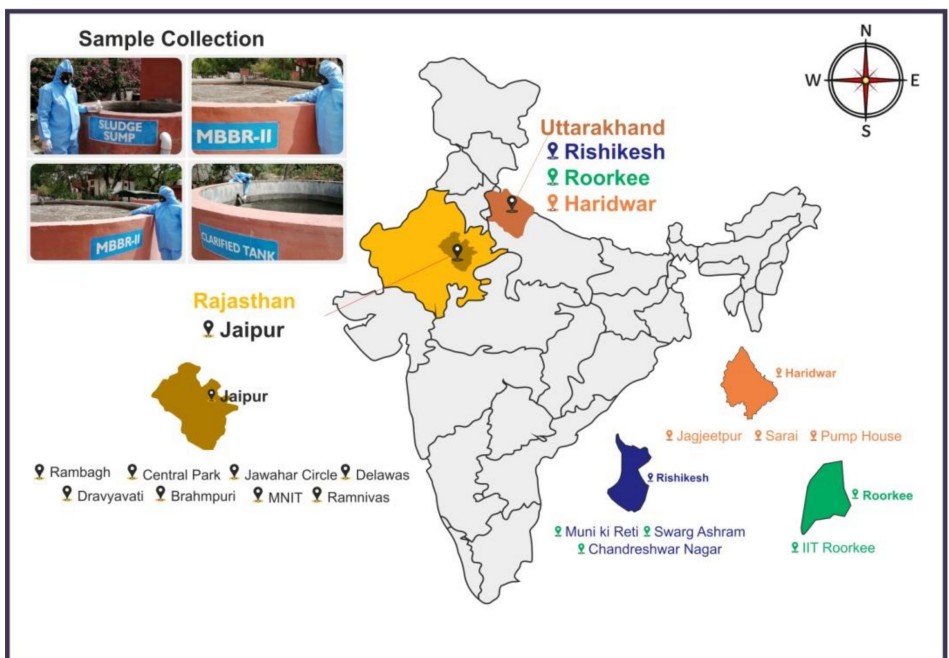

**Figure 1.** Locations of samples collected from the Uttarakhand and Rajasthan states of India.

### 2.2. Sample Pre-Processing

The samples were pre-processed using two different methods, as specified in Figure 2. In method A, a 50 mL sample was transferred into sterile falcon tubes in the Biosafety Cabinet (BSL-II), followed by surface sterilization of the falcons using 70% ethanol and UV light exposure for 30 min. The heat inactivation of the virus was then achieved by placing the falcon tubes in a water bath at 60 °C for 90 min. The samples were further filtered through a 0.45μm membrane using a vacuum filter assembly. The filtrate was then transferred to a fresh falcon containing 4 g PEG (Himedia) and 0.9 g NaCl (Himedia). The content was dissolved through manual mixing, followed by centrifugation at 4 °C for 30 min at $7400\times g$. The pellet obtained was then resuspended in 1X Phosphate Buffer Saline (PBS). The method B used for the detection of the SARS-CoV-2 virus was performed by the transfer of a 1 mL sample in a 1.7 mL centrifuge tube, followed by centrifugation at $7400\times g$ for 15 min. The supernatant was collected in a fresh tube and was again centrifuged at 7000 rpm for 15 min. The supernatant thus obtained was used for nucleic acid extraction. The samples for the duration of May and June were pre-processed using

method A. Because method B requires a shorter duration (2 h 40 min), as compared to method A (as described in Figure 2), and because both methods gave a similar efficiency of detection (as reported in our previous study, Arora et al. [27], all samples from July 2020 onwards were pre-processed using method B.

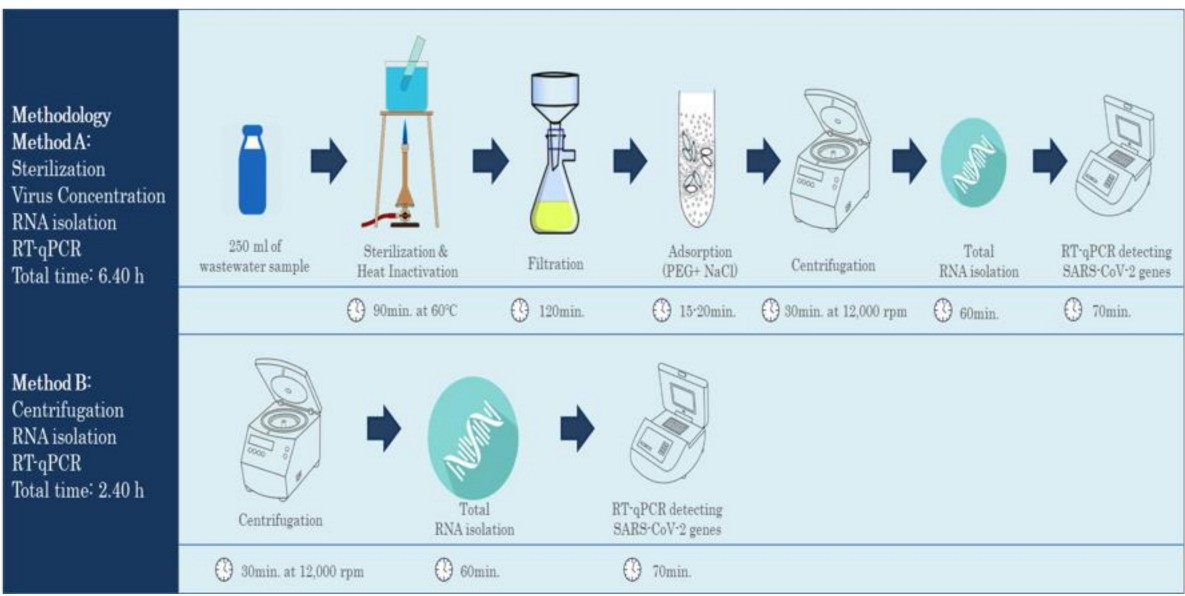

**Figure 2.** Methodology for the detection of SARS-CoV-2 viral genome through RT-qPCR in wastewater samples.

### 2.3. RNA Isolation

RNA extraction and the subsequent steps of detection were completed at the Dr. B. Lal Clinical Laboratory Pvt. Ltd., Jaipur (which is authorized by ICMR to conduct COVID-19 testing in humans). The viral RNA molecules present in the wastewater samples from May to early July 2020 were isolated using a Biospin kit (Cat# BSC77M1). As per the vendor's instructions, 10 μL proteinase K and 200 μL of lysis buffer were added to 200 μL of the sample into a 1.5 mL centrifuge tube, followed by vortex mixing and incubation at 56 °C for 15 min in a heating block. 250 μL of ethanol was then added to the sample and mixed by vortexing for 15 s. The mixture was then transferred to the spin column and centrifuged at $10,000 \times g$, followed by sequential washing with the three wash buffers provided in the kit, followed by centrifugation at $10,000 \times g$ for 1 min at each washing step. After the complete drying of the spin column, the RNA was eluted out using a 50–100 μL elution buffer. Centrifugation was done at $12,000 \times g$ for 1 min. The RNA from the samples collected in late July to August was extracted using the automated Kingfisher Flex System ™ (Cat#5400610).

### 2.4. Real-Time PCR for Detection of SARS-CoV-2

For the qualitative detection of SARS-CoV-2 in the wastewater samples, RT-qPCR was performed using a commercially available FDA-approved Allplex™ 2019-nCoV Assay kit (Cat# RP10244Y, 208 RP10243X), as per the vendor's instructions for the qualitative detection of SARS-CoV-2 genomic RNA in the sample on Applied Biosystems™ QuantStudio™ 5. The master mix was prepared using the kit content, which was composed of an amplification and detection reagent, enzyme mix for one-time RT-qPCR, buffer containing dNTPs, buffer for one-step PCR and RNase-free water. Each PCR tube contained 8 μL sample RNA, 5 μL 2019-nCoV oligo -mixture, 5 μL Real-time one-step buffer and 2 μL Real-time One-step enzyme and the final volume of the mixture were adjusted to 25 μL using RNase free water. Additionally, as per the kit protocol, "no template" as a negative control, assay target gene plasmids as positive controls, and MS2 phage DNA as internal controls were used to validate each round of reactions. Three genes (E, RdRp, and N)

were targeted to detect the presence of the SARS-CoV-2 genome. A list of the different fluorophores used for the detection run is given in Supplementary Materials. Thermal cycling reactions were performed at 50 °C for 20 min, 95 °C for 15 min, 44 cycles at 94 °C for 15 s, and 45 cycles at 58 °C for 30 s, in a thermal cycler. The Allplex™ 2019-nCoV Assay is an in vitro diagnostic real-time reverse transcriptase-polymerase chain reaction (RT-PCR) test used for the qualitative detection of SARS-CoV-2 viral nucleic acids. The kit has reagents RP-V IC (1000 µL), composed of MS2 phage genome as an exogenous internal control. As per the manufacturer's instructions, internal control Ct values above 40 are considered invalid. The PCR controls are provided with the Allplex™ 2019-nCoV Assay to confirm the validity of each PCR run on the same plate. Negative Control (NC) is used as a PCR control to confirm test validity and the absence of any contaminants during testing. The "no template" control is prepared using RNase-free water added to Master Mix prior to PCR. NC was included in each test run.The PC is constructed using plasmids encoding Allplex™ 2019-nCoV Assay target sequences and was included in each test run.

The criteria, of choosing two out of three genes with valid Ct values as criterion for overall positive or not, is based on manufacturer's instruction. This has been extensively discussed in Kumar et al. [21,22], bas well. In addition, it is important to note that, as we detect titer of RNA and talk about the entire SARS-CoV-2 genome, it would not be rational to say that the entire genome exists based on just one gene detection. Therefore, at least two out of three genes need to be present in a sample in order for the sample to be announced as positive.

## 3. Results and Discussions

### 3.1. Detection of SARS-CoV-2 RNA in Untreated Wastewater Samples

In India, the country-wide lockdown and prohibitions pertaining to the containment of the pandemic were enforced on 24 March 2020. This was initially declared as a 21-day restriction period, but was extended multiple times until 31 May 2020. With the onset of the post-lockdown period, most of the states lifted restrictions and prohibitions, bar a few precautionary measures. By July 31, while the rate of the country's confirmed positive cases was constantly rising [28], many of the states were completely out of the lockdown conditions. The window of wastewater influent sample collection in the present study started as early as 4 May 2020 and continued throughout the lockdown and until post-lockdown, on 14 August 2020. The observations discussed in Table 3 are taken from sampling completed at various sites in Rajasthan and Uttarakhand and tested for the presence of the viral genome. The early sampling in Rajasthan coincided with the onset of community restrictions and the lockdown duration and continued through the gradual relaxation of the lockdown. The sampling was carried out until August 2020, and coincided with the time window when most establishments, including offices, barbershops, markets, malls, were opened and unrestricted city movements were allowed. In Uttarakhand, sampling was conducted under partial lockdown conditions, with offices closed and restricted activities permitted in some areas.

The wastewater of the entire Jaipur city, and the outskirts area is connected through a sewerage network that joins the main trunk sewer with a 1800 mm diameter and an average flow of 130 MLD, terminating at Delawas Sewage treatment plant (STP) Site 3, based on the activated sludge process (centralized treatment facility at 125 MLD capacity). From this main trunk, settled sewage is withdrawn at a rate of 1 MLD Sites 1, 2, and 4, has been connected to three decentralized treatment plants based on MBBR technology for more than ten years in order to maintain public parks. Besides this, Site 6 has its own WWTP plant at MNIT Jaipur campus area, based on MBBR technology of a capacity of 1 MLD. Site 7 at Dravyavati River WTWP has a capacity of 65 MLD. Site Brahmpuri WWTP (Site 5) has a capacity of 27 MLD. Almost all of the WWTPs are working at full capacity, and hence this can be considered to be their average flow rates. The wastewater from the majority of the Haridwar city gets collected to the pump house (Site 14), and then travels to two major WWTPs at Site 12 and 13, which are SBR and MBBR based, respectively. These

WWTPs are not working in their full capacity, and their average flow rate is provided in Table 2.

**Table 3.** Results showing Ct values of targeted genes for detection of SARS-CoV-2 RNA in untreated and treated wastewater samples collected from different WWTPs.

| | | | | | | | | |
|---|---|---|---|---|---|---|---|
| **(A) Rajasthan** | | | | | | | |
| **Site No.** | **Sampling Site** | **Type of Sample** | **Sampling Date** | $C_{TE}$ | $C_{TR}$ | $C_{TN}$ | **Final Result Interpretation** |
| Site 1 | MBBR, Ramniwas Garden, Jaipur | Untreated | 4 May 2020 | 33 | 36 | 33 | Positive |
| | | | 15 May 2020 | - | - | - | Negative |
| | | | 20 May 2020 | 31 | 38 | 34 | Positive |
| | | | 12 June 2020 | 32 | 37 | 34 | Positive |
| | | | 12 July 2020 | 36 | - | 36 | Positive |
| | | | 11 August 2020 | 35 | 36 | 36 | Positive |
| | | Secondary treated | 11 August 2020 | - | - | - | Negative |
| | | Tertiary treated | 4 May 2020 | - | - | - | Negative |
| | | | 12 June 2020 | 35 | - | - | Negative |
| | | | 26 July 2020 | - | - | - | Negative |
| | | | 11 August 2020 | - | - | - | Negative |
| Site 2 | SBR, Central Park, Jaipur | Untreated | 4 May 2020 | - | - | - | Negative |
| | | | 15 May 2020 | - | - | - | Negative |
| | | | 20 May 2020 | - | - | - | Negative |
| | | Secondary Treated | 11 August 2020 | - | - | - | Negative |
| | | Tertiary treated | 4 May 2020 | - | - | - | Negative |
| | | | 11 August 2020 | - | - | - | Negative |
| Site 3 | ASP, Delawas, Jaipur | Untreated | 4 August 2020 | 32 | 36 | 36 | Positive |
| | | | 8 August 2020 | 34 | 35 | 36 | Positive |
| | | Secondary treated | 11 August 2020 | - | - | - | Negative |
| Site 4 | MBBR, Jawahar Circle, Jaipur | Untreated | 4 May 2020 | - | - | - | Negative |
| | | | 12 June 2020 | - | - | - | Negative |
| | | | 4 August 2020 | 34 | - | 38 | Positive |
| | | | 8 August 2020 | - | - | - | Negative |
| | | | 11 August 2020 | - | - | - | Negative |
| | | | 29 October 2020 | 34 | - | 33 | Positive |
| | | Primary treated sample | 29 October 2020 | 31 | - | 32 | Positive |
| | | Secondary treated | 8 August 2020 | - | - | - | Negative |
| | | Tertiary treated | 8 August 2020 | - | - | - | Negative |
| Site 5 | SBR, Brahmpuri, Jaipur | Untreated | 15 May 2020 | - | - | - | Negative |
| | | | 20 May 2020 | 36 | - | 37 | Positive |
| | | | 12 June 2020 | 36 | - | 36 | Positive |
| | | | 11 August 2020 | 33 | 34 | 35 | Positive |
| | | Secondary treated | 12 June 2020 | - | - | - | Negative |
| | | | 11 August 2020 | - | - | - | Negative |
| Site 6 | MBBR, MNIT, Jaipur | Untreated | 4 May 2020 | - | - | - | Inconclusive |
| | | | 4 August 2020 | - | - | - | Negative |
| | | Tertiary treated | 4 May 2020 | - | - | - | Negative |
| Site 7 | SBR, Dravyavati River Project, Jaipur | Untreated | 4 May 2020 | – | - | - | Negative |
| | | Tertiary treated | 4 May 2020 | - | - | - | Negative |

**Table 3.** *Cont.*

| | | | | $C_{TE}$ | $C_{TR}$ | $C_{TN}$ | |
|---|---|---|---|---|---|---|---|
| **(B) Uttarakhand** | | | | | | | |
| Site 8 | SBR, IIT Roorkee | Untreated | 25 July 2020 | 33 | 37 | 37 | Positive |
| | | Untreated | 14 August 2020 | 36 | 30 | 37 | Positive |
| | | Secondary treated | 11 August 2020 | - | - | - | Negative |
| | | Tertiary treated | 11 August 2020 | - | - | - | Negative |
| Site 9 | MBBR, Muni kiReti, Rishkesh | Untreated | 25 July 2020 | 36 | 37 | 37 | Positive |
| | | Tertiary treated | 25 July 2020 | - | - | - | Negative |
| Site 10 | SBR, Swarg Ashram, Rishikesh | Untreated | 25 July 2020 | - | - | - | Negative |
| Site 11 | MBBR, Chandreshwar Nagar, Rishikesh | Untreated | 25 July 2020 | - | - | - | Negative |
| | | Tertiary treated | 30 July 2020 | - | - | - | Negative |
| Site 12 | SBR, Sarai, Haridwar | Untreated | 25 July 2020 | 36 | - | 37 | Positive |
| | | Tertiary treated | 25 July 2020 | - | - | - | Negative |
| Site 13 | MBBR, Jagjeetpur, Haridwar | Untreated | 25 July 2020 | 33 | 37 | - | Positive |
| | | Tertiary treated | 25 July 2020 | - | - | - | Negative |
| Site 14 | Sewage, Pump House, Haridwar | Untreated | 25 July 2020 | - | - | 37 | Negative |

$C_{TE} = C_T$ value of E gene, $C_{TR} = C_T$ value of RdRp gene, $C_{TN} = C_T$ value of N gene, $C_{TIC} = C_T$ value of Internal Control. The value of Ct above 40 indicates that the gene tested is not present in the sample. The presence of at least two out of three positive genes in a sample was ruled to be positive for the SARS-CoV-2 genome.

The observations from the RT-qPCR-based qualitative genome detection showed that the spread of COVID-19 in Northern India is highly extensive. Table 3 shows the results of the untreated and treated wastewater samples, and the Ct values of all three target genes (E, RdRp, and N) were interpreted accordingly. A total of 40 samples from Rajasthan and 14 samples from Uttarakhand collected between May 4 and August 14 (2020) were tested for the presence of SARS-CoV-2 RNA. Some variations in Ct values are probably due to the complexity and variability of the sewage samples and has been described before [29,30]. The interpretation of results in Table 3 (from different sites, as described in Table 1) was based on categorizing a sample as positive when the cycle threshold took place below cycle 40, for either two of the three genes such as RNA-dependent polymerase (RdRP), nucleocapsid (N) SARS-CoV-2 specific genes and gene E, characteristic of pan-Sarbeco viruses detection, as per manufacturer's instructions. The Ct values of positive samples were in the range of 30–38, corresponding to a mild to moderate genome load presence in all of the untreated wastewater samples. It was observed that the areas served by the WWTPs of Jaipur city that showed positive results reported a continuous increase in confirmed positive patients, which corroborated with the Ct values. As observed in Table 2, during May 2020, the samples showed the presence of SARS-CoV-2 RNA in untreated wastewater from Site 1 and Site 5, which corroborated with the positive patients' cases around the area (as reported in [27], while the rest of the sites showed negative samples. With the withdrawal of lockdown restrictions by the end of June 2020, a significant increase in the COVID positive cases was observed. The number of positive cases reached the thousands, as reported by [4]. This can be corroborated with the decrease in the Ct values from >40 to 33, which can be seen for certain sites such as Site 5, Brahmpuri, Jaipur. However, the results obtained from the Site 1 samples are quite captivating. The Ramniwas Garden WWTP (Site 1) is currently serving the walled city area of Jaipur that includes the major hotspot of the city, the Ramganj area, having the maximum reported cases in May 2020. The detected genome load in the sample collected from Site 1 increased during the lockdown, and decreased in the post-lockdown period. One of the key reasons behind this observation is that the SMS Hospital sewage is also collected and treated at Site 1 WWTP. SMS Hospital had a COVID-19 positive patient load in May 2020 (during lockdown in the city), and the load gradually decreased in July and August. This is because patients started to recover, and there was also a decline in positive patient admissions in the hospital. Therefore, this pattern of Ct values and infectivity during and post lockdown periods can be explained.

The Ct values in the wastewater samples collected from Uttarakhand indicate genome loads within the same order as the loads present in Jaipur during June 2020. This was during the time when restrictions were being partially lifted in the cities. The increase in the viral genome load concurs with the gradual rise in the number of infected individuals, which rose from 2138 and 720 active cases on June 1 to 10,260 and 5912 active cases by the end of August 2020 in Jaipur and Haridwar districts, respectively (number of cases for Jaipur district were obtained from the newspaper, the case numbers for Haridwar were reported by the Department of Medical Health and Family Welfare [31] (https://health.uk.gov.in/, accessed on 31 August 2020).

A few studies have also tried to correlate the Ct values with the genome load and the probability of that load being infective [32,33]. Different lab-scale studies were conducted investigating the percentage of the cell cultures turning positive at various Ct levels of the SARS-CoV-2 genome detected in the clinical samples of sputum and nasal swabs [32]. The study shows that a sample with a genome load with Ct values greater or equal to 34 could not infect the cell lines tested and postulates that the patients with higher or equal to 34 Ct values may be discharged [33]. However, these studies have been conducted in labs for clinical samples and might not correspond to the infection probabilities that could occur through the wastewater contamination. Thus, it would be interesting to investigate the possibilities of transmission routes through contaminated wastewater [34].

It was further observed that gene E was frequently detected with the lowest Ct values during the qualitative detection compared to the other two genes (Genes N and RdRp) (as reported in Table 3 and Figure S1). In Uttarakhand, the Ct of E gene in four samples out of five positive samples was the lowest compared to those of N and RdRp genes in the same samples. In Jaipur, all the five-times sampling in August 2020 (overall ten out of twelve samples), showing positive results for the presence of the SARS-CoV-2 genome, had the lowest Ct values for the E gene. Furthermore, it was observed that for the samples collected during the earlier time points from Jaipur, the Ct values of E and N indicated a marginal difference in their respective gene loads (Figure S1a). In contrast, the sample collected later in the time window from July to August clearly showed the prevalence of the E gene over the other two genes (as depicted in Figure S1b–d), which was reflected by the rise in the graph E gene for fewer samples.

Thus, the present study highlights the effect of lifting the lockdown restrictions with the increase in the viral genome load per unit of wastewater. In the post lockdown period (August 2020), the rapid increase in the numbers of COVID-19 patients was corroborated by the decrease in Ct values (Site 5). Additionally, the genes tested for SARS-CoV-2 in the wastewater showed different gene load levels, as indicated by the Ct values. The E gene seems to be present more abundantly than N and RdRp in all of the samples. Either of two reasons can explain this observation. One reason could be related to the host-pathogen interaction, which is different for different populations. Thus, it is possible that a particular gene is more abundantly or stably expressed in a community. Another reason could be that the E gene is responsible for the structural assembly of the viral particles without interacting with N and RdRp genes [35].

Additionally, the self-assembly of SARS-CoV-2 requires the interaction of the N gene with viral RNA for compaction and packaging into the viral capsid [36]. Thus, the observed Ct value trends might be interesting and may indicate shedding of viral capsids before the packaging is complete. Alternatively, the trends of Ct values could indicate the differential expression rates of these genes under different conditions, such as the genetic makeup of a community, geographical, climatic, etc. Therefore, these dynamics in E, N, and RdRp gene detection might prove useful in understanding the viral host interactions and transmission probabilities through wastewater.

The study also highlights the methods of transporting the wastewater samples, and its effect on RNA detection. The samples from the state of Uttarakhand were collected and immediately transported to Jaipur at 4 °C using cold chain transportation. Despite the gap of approximately 3–4 days between sampling and pre-processing, the Ct values

observed for these samples indicated a mild to moderate range of genome load, similar to the immediately pre-processed samples. This indicates that transportation before pre-processing did not significantly affect the detection and that transportation of the collected wastewater samples at 4 °C might be a sufficient measure for genome detection. This observation is important in the context of using only qualitative detection of SARS-CoV-2 in wastewater, which was to be transported over a longer distance using cold chain transportation. However, in the case of quantification, more studies are required in order to understand this.

*3.2. Secondary Aerobic Biological Treatments Are Sufficient to Decay the SARS-CoV-2 RNA beyond Detection*

The presence of the SARS-CoV-2 genome in untreated wastewater is a cause forconcern as the wastewater is a potential route of viral transmission to sanitation workers. Additionally, aerosolization of wastewater during its treatment can promote infection via air, provided that the viral particle is active. The sludge and treated water from these treatment facilities are used for agricultural purposes, which can put end users' health at risk. To investigate the probability of such a transmission route, samples from various stages of aerobic wastewater treatment were collected and checked for the presence of viral RNA.

The samples were collected from primary treatment, secondary treatment, and tertiary treatment stages of wastewater treatment plants. Different treatment methods were available at different sites, which provided comprehensive information on the fate of viral RNA after different methods of treating the wastewater had been implemented. While most of the sites selected in Jaipur have only secondary treatment technology, three of them (viz. Site 1 Ramniwas Garden WWTP, Site 4 Jawahar Circle WWTP, and Site 6 MNIT WWTP) have tertiary treatment technology, with a UV disinfection unit. However, during our test window, only Site 1, Ramniwas Garden WWTP, was connected to a community that was considered to be a hotspot for the pandemic, while the other two sites were not. In order to investigate the effect of each wastewater treatment stage on SARS-CoV-2 decay, samples from each of the stages mentioned above were checked. The sites which were not serving the hotspot communities showed the absence of viral RNA in influent and samples from the subsequent stages. However, it was observed that even where we could detect the presence of viral RNA in the influent, the viral RNA decayed beneath the level of detection immediately after the second stage treatment (Table 3) and remained undetectable at consequent stages as well. In fact, the detectable and intact SARS-CoV-2 viral genome was not observed in any of the post-secondary treated wastewater samples, regardless of the type of biological treatment (i.e., Activated sludge process, MBBR or SBR). The absence of viral RNA in the effluent was consistent between the sampling that was conducted during the lockdown and after the lifting of the lockdown regulations. While it is important to note that there has been a constant increase in the number of COVID-19 cases, and by extension, the SARS-CoV-2 genome load per sample unit, treated samples were still consistently negative for the viral genome presence.

The observations thus far indicated that the viral RNA was decaying between the stages of influent and post-secondary treatment. These observations raised questions about which particular treatment could be directly involved in the decay. Any of three possible reasons could be the source of the decay: firstly, that the viral load was going down during the primary treatment procedures itself due to the settling down of the viral particles; secondly, it was very likely that the secondary treatment procedures, which are effective in virus removal by the biofilm generated by the microbes, were responsible; or thirdly, it was possible that both the primary and secondary treatments together were responsible for the decay with a partial depreciation in load starting at the primary treatment stage itself. Samples were taken and analyzed from untreated wastewater and during all the subsequent stages of treatment in order to delineate between these mechanisms. The Ct values of all target genes obtained between untreated and post-primary treated wastewater samples showed little variation, while most of the viral RNA was decayed during the

secondary treatment stage. This indicates the possibility that the second mechanism, where the decay of RNA happens due to the biological treatments, is in action. This evidence thus shows that the wastewater treatment facilities are capable of degrading the viral RNA significantly. The biological treatment stages were capable of completely degrading the intact SARS-CoV-2 genome beyond the detection sensitivity and this did not depend on the tertiary treatment (i.e., disinfection stage). Direct chlorination of untreated sewage might not significantly reduce the detected viral loads if the chlorine demand of the sample is not satisfied. Chlorine demand is directly proportional to the organic waste matter present in the water samples [37]. Since there is a very large quantity of organic matter in the sewage, it is understandable if chlorination alone is not highly effective. Evidently, Zhang et al. [20] found an unexpected occurrence of SARS-CoV-2 viral RNA in aseptic tank even after disinfection with sodium hypochlorite. They suggested reevaluation of the existing disinfection approach (free chlorine: >6.5 mg/L after 1.5-h contact).

In contrast, the absence of any detectable viral genome in wastewater samples collected post-secondary treatment might indicate the efficacy of the biofilm generated by the microflora in the biological reactors in removing the viral genome loads. This hypothesis is based on several studies that have reported biofilms' role in the removal of various types of viruses [38,39]. A biofilm can be defined as a well-organized community consisting of cooperating microorganisms immobilized in an extracellular polysaccharide (EPS) matrix [40,41]. Biofilms can be an association of single or multiple species of bacteria, fungi, algae, protozoans, and rotifers in combinations [41]. Thus, it is possible that these biotic constituents of the biofilm, along with the abiotic components, such aspH, temperature, or minerals present, are an integral part of the SARS-CoV-2 viral decay and removal from the wastewater. Further investigation into the role of biofilm is much needed, as the aerobic biological wastewater treatment process investigated in this study seems to be more efficient in the decay of viral RNA than the anaerobic UASB system, in which the decay is completed only after the post-secondary treatment aeration stage [20].The findings from the present study indicate that secondary aerobic biological WWTPs contribute to reducing the virus concentration due to the adverse environmental conditions (i.e., temperature, solids, pH, or disinfectants) to make the water fit for reuse.

### 3.3. Comparison of the Efficiency of Studied Treatment Processes

We examined the efficiency of MBBR, SBR, and ASP wastewater treatment processes by comparing the changes in the Ct values of the SARS-CoV-2 E gene, RdRp gene, and N gene before and after the treatment wastewater samples (i.e., influent and effluent). A paired-samples T-test was carried out in order to evaluate the efficacy of MBBR and SBR treatment processes (Figure 3a,b), while a comparison was made between influent and effluent wastewater samples of the ASP treatment process (Figure 3c). The results showed significant removal of all three targeted SARS-CoV-2 genes from the MBBR plant ($p < 0.05$), while a substantial decrease ($p > 0.05$) in E and N genes was noticed in the SBR treatment process depicted by a post-treatment increase in Ct values of genes (Figure 3a,b). Likewise, all three genes were successfully removed from the ASP treatment process (Figure 3c).

In addition to this, the paired T-test between the inlet and outlet wastewater samples, taken on the same date during the study, displayed a significant reduction/removal of SARS-CoV-2 genes, except for one occasion in the case of MBBR treatment (Figure 4). Contrary to this, the reduction of SARS-CoV-2 genes in wastewater samples was insignificant in the SBR treatment process. The results suggest that all three treatment processes successfully reduced/removed the virus genetic load in wastewater samples; however, the performance of MBBR was found to be higher than that of the SBR and ASP treatment processes.

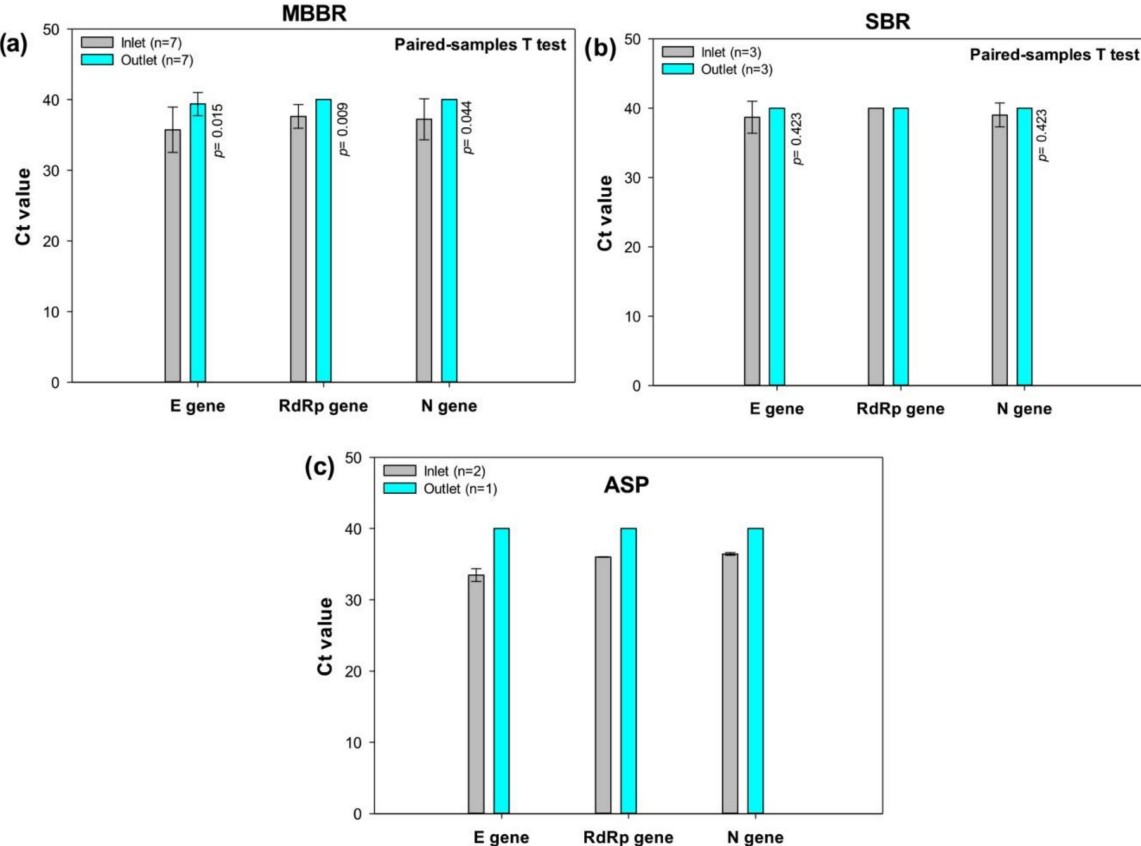

**Figure 3.** Comparison between the inlet and outlet wastewater samples for SARS-CoV-2 genetic load in (**a**) Moving bed Bioreactor (MBBR) based treatment; (**b**) Sequencing batch reactors (SBR); and (**c**) Activated Sludge Process (ASP).

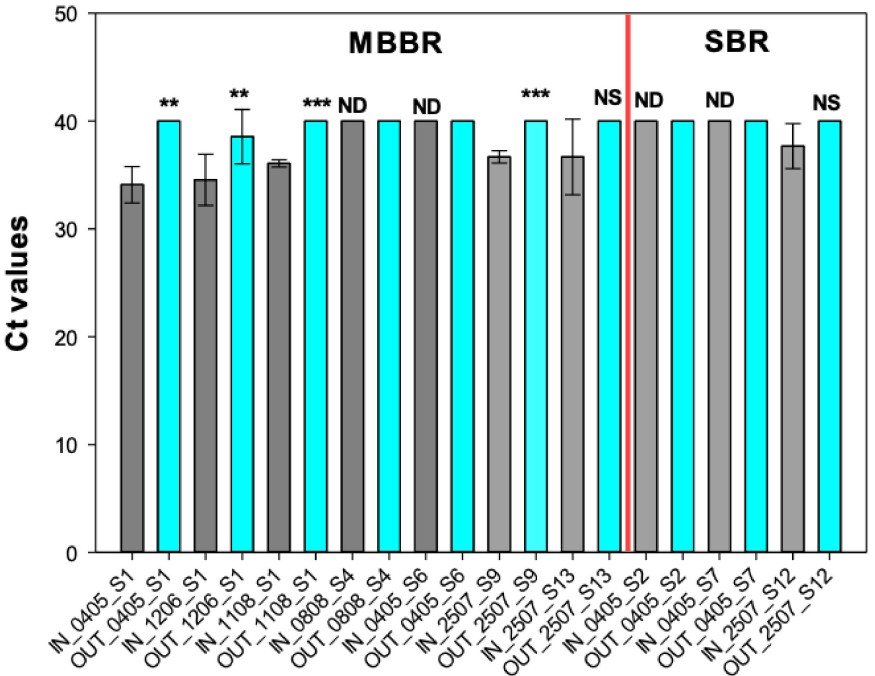

**Figure 4.** Paired T-test between the inlet and outlet wastewater samples, taken on the same date for SARS-CoV-2 genetic load in Moving bed Bioreactor (MBBR) based treatment and Sequencing batch reactors (SBR). where *** = $p < 0.01$; ** = $p < 0.05$; NS = not significant; ND = not detected; and RT-PCR was run for 40 cycles).

## 4. Conclusions

The present study reports 33.3 percentage of positive wastewater samples for SARS-CoV-2 from two states of Northern India, with none of the secondary or tertiary treated samples found to be positive. The samples from multiple locations make the study more representative and indicate the potential applications of WBE across diverse geographical and climatic conditions. This highlights the fact that aerobic biological wastewater treatment systems might be sufficient for SARS-CoV-2 removal, diminishing any possibility of the fecal route of disease transmission through treated wastewater. This study also assessed the efficacy of each stage of aerobic wastewater treatment systems and established a surveillance system through sewage monitoring of the potential virus circulation. This study further highlights that Ct values correspond to positive patient cases in the area and illustrated the effect of physical distancing and lockdown regulations, as is evident from several negative samples during the lockdown period. The paired T-test between the inlet and outlet wastewater samples, taken on the same date during the study, displayed a significant reduction/removal of SARS-CoV-2 genes. The results suggest that all three treatment processes successfully reduced/removed the virus genetic load in wastewater samples; however, the performance of MBBR was found to be higher than that of the SBR and ASP treatment processes. This study opens up a new direction of treatment efficacy on SARS-CoV-2 removal and stresses the need to understand the survival of SARS-CoV-2 under natural conditions in various aerobic wastewater treatment systems. The present study aims to add to the existing literature on WBE and contribute to an efficient and resilient public health emergency response mechanism in India for the future. This study provides a comprehensive data analysis and gives insights into the role of aerobic biological treatment systems in decaying the SARS-CoV-2 viral genome.

**Supplementary Materials:** The following are available online at https://www.mdpi.com/article/10.3390/w13162265/s1, Figure S1: Graphs showing trends in the three genes from the months of May to August for (a) May 20 (b) June 12 (c) July 25 (d) August 4 (e) August 11 (2020).

**Author Contributions:** Conceptualization, S.A., A.N., A.B.G.; methodology, A.N., J.S., J.R.; validation, V.K.T., M.K., formal analysis, A.B.G., A.A.K., A.R.; investigation, A.N., D.S., J.S., J.R., resources, S.A., S.K.S., S.S.; data curation, S.B.T., V.S., M.K.; writing—original draft preparation, A.N.; writing—review and editing, S.A., V.K.T., S.B.T., V.S., M.K.; visualization, V.K.T., supervision, A.B.G.; project administration, S.A.; funding acquisition, V.K.T., A.A.K. All authors have read and agreed to the published version of the manuscript.

**Funding:** This research was funded by research grants from the Department of Biotechnology-GoI [Grant No. BT/RLF/Re-entry/12/2016].

**Institutional Review Board Statement:** Not applicable.

**Informed Consent Statement:** Not applicable.

**Data Availability Statement:** Not applicable.

**Acknowledgments:** We would like to acknowledge the constant support received from B. Lal Gupta (Director) and Aparna Datta (principal) for providing support. We would also like to thank the Centre for Innovation, Research & development (CIRD, Dr. B. Lal Clinical laboratory Pvt. Ltd.) for their support in analysis. We would also thank the Jaipur development authority (JDA) officials & plant operators at WWTPs and all other participating wastewater treatment plants for the collection of sewage samples in this study.

**Conflicts of Interest:** The authors declare no conflict of interest.

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
