# Peer review of "Imprints of Lockdown and Treatment Processes on the Wastewater Surveillance of SARS-CoV-2: A Curious Case of Fourteen Plants in Northern India"

_water, doi:10.3390/w13162265_

Round 1
Reviewer 1 Report
Overall, I enjoyed reading this paper by Arora et al. It nicely describes the impact of lockdown and wastewater treatment processes on SARS-CoV-2 detection in wastewater samples in 14 treatment plants in Northern India. The introduction, methods and results as well as interpretation of the findings are sound. This paper represents an important addition to the wastewater literature, particularly in India where wastewater surveillance of SARS-CoV-2 is starting to be recognised as an exceptionally useful epidemiological surveillance tool.
I have a few minor suggestions:
Line 29 – spelling of coronavirus (all one word)
Lines 33-34 – more information on the districts; which states?
Lines 35-36; change to correlation of SARS-CoV-2 RNA with incident inpatient and community cases of COVID-19
Line 45 - state when outbreak emerged in Wuhan
Line 70 - remove brackets around Wurtzer
Line 105 - remove 'h' after Uttarakhand
Univer section 2.2 - please convert rpm to g
Lines 158-159; please amend degrees Celsius with appropriate superscript
Line 203 – separate out words some variation in..
Line 368 - please remove 'were' that comes before successfully reduced
Reviewer 2 Report
In the this form the novelty of the paper is hard to find, and in any case it must be better highlighted. Without this clarification, it is difficult for me to recommend the manuscript for publication in its present form.
ABSTRACT
The abstract should be concise and specific and consequently should be revised. The abstract should provide background knowledge focused on the topic to be addressed, state the objectives of the study.
Authors should summarize the central core of knowledge that is the focus of the paper and better discuss the importance and relevance of their main findings.
INTRODUCTION
Authors should be reenforce the introduction with environmental concentration, current state of the art as well as gaps-of-knowledge and not repeat the abstract
CONCLUSIONS
Conclusions should be reenforced considering the main findings of the study.
Hope my comments will help authors to improve their manuscript!!!
